# Preschool Children’s Physical Activity and Community Environment: A Cross-Sectional Study of Two Cities in China

**DOI:** 10.3390/ijerph192214797

**Published:** 2022-11-10

**Authors:** Yu Wang, Gang He, Kaiyue Ma, Dongsheng Li, Chao Wang

**Affiliations:** School of Kinesiology and Health, Capital University of Physical Education and Sports, Beijing 100191, China

**Keywords:** physical activity, accelerometer, built environment, preschool children

## Abstract

Research on the relationship between preschool children’s physical activity (PA) and community environment is limited and inconclusive, yet understanding this relationship is important to acquire sufficient information to guide the development of intervention programs. This study aims to objectively measure preschool children’s PA and examine associations between PA and the community environments. A total of 471 preschool children aged 3–6 years old were recruited from eight kindergartens in Beijing and Zhengzhou. PA was measured by accelerometers. Light PA (LPA), moderate PA (MPA), and vigorous PA (VPA) were computed on the basis of cutoff points developed for preschool children. Moderate-to-vigorous PA (MVPA), and total PA (TPA) were obtained by calculation. Children’s active transportation modes were indicated by the frequency of active trips (FAT) reported by parents. The community environment was collected by parental scales. Multivariate linear regression was used to analyze the associations between PA and the community environment. In total, 304 preschool children (mean age 5.07 ± 0.94 years, 50.66% boys) were included in the final analysis. Children spent an average of 77.58 ± 18.78 min/day in MVPA and 173.26 ± 30.38 min/day in TPA. Linear regression showed that ‘parental perception of appropriate walking distance’ was associated with nearly half of the indicators of the children’s PA. ‘Public activity facilities near the community’ was associated with FAT for overall children (*B* = 0.099, 95% CI = 0.014, 0.183). ‘Community transportation environment’ was associated with overall children’s average day LPA (*B* = 4.034, 95% CI = 0.012, 8.056), weekend LPA (*B* = 8.278, 95% CI = 1.900, 14.657), MPA (*B* = 4.485, 95% CI = 0.613, 8.357), TPA (*B* = 14.777, 95% CI = 2.130, 27.424), and FAT for girls (*B* = −0.223, 95% CI = −0.443, −0.003). Furthermore, ‘community personal safety’ was associated with boys’ weekday VPA (*B* = −3.012, 95% CI = −5.946, 0.079). Parental perception of appropriate walking distances, improvement of PA facilities, community personal safety, and the community transportation environment all contribute to the prevention of preschool children’s PA patterns deterioration.

## 1. Introduction

Childhood obesity has become a growing public health problem as rates of overweight and obesity have risen globally over the past three decades [1], and the incidence of overweight and obesity among preschool children in China has been on the rise in recent years [2,3]. One of the factors associated with raising rates of overweight and obesity in children is a decrease in physical activity (PA) [4]. Regular PA in early childhood has many benefits, such as improved cardiometabolic health, musculoskeletal development, and psychosocial health [5,6]. Despite all the benefits of PA, the status quo for preschool children in China is not promising [7,8]. Preschool is a crucial time for growth and development, during which children should be physically active and cultivate an active lifestyle to prevent obesity and reduce their risk of chronic disease in adulthood [9,10,11].

Effective promotion of PA in preschool children presupposes a full understanding of its influencing factors. According to social-ecological theory, an appropriate environment (including family, community, social, policy, etc.) is required to shape healthy behaviors (e.g., PA participation) [12]. In China, the community is the basic unit of urban space and the first environment for young children to exercise their activity rights [13].

To date, a few studies confirm the relationship between preschool children’s PA and their community environment [14,15]. However, the findings of the existing studies are inconsistent and inconclusive. For example, a satellite image-based study in the United States shows that the greener the community, the higher the level of outdoor activity among preschool children [14]. However, Michael’s accelerometer study finds no significant correlation between objective community environmental factors (such as social status at the community level and the proportion of forests, recreation, and residential areas) and PA among preschool children [15]. Behind the motorization of western cities is the spread of low-density cities, but the overall planning of Chinese cities has long encouraged the mixed use of land functions, so China has formed a high-density and mixed urban form before motorization [16]. Research on the community environment and PA of preschool children in China is very limited though [17]. Therefore, the relationship between PA and community environmental factors in preschool children in China should be investigated to acquire sufficient information to guide intervention strategies for promoting preschool children’s PA from the perspective of urban planning.

The purposes of this study are to objectively measure preschool children’s PA and explore the relationship between PA and the community environment. In so doing, this study can provide a reference for relevant departments to take measures to promote children’s PA and improve the current situation of children’s physical inactivity.

## 2. Materials and Methods

### 2.1. Participants

A total of 471 preschool children between the ages of 3 and 6 years were recruited from eight kindergartens in Beijing and Zhengzhou, two of the most population-dense cities in China. Participants had to be in good health, physically fit, and able to engage in regular PA. Their parents were given informed consent forms and details of the study, from which participants could opt out at any time during the test. This study was approved by the Ethics Committee of Capital University of Physical Education and Sports.

### 2.2. Physical Activity

Participants’ PA consisted of the following two main components: PA time and active transportation modes. PA time was measured by the ActigraphGT3X+ accelerometer (ActiGraph, LLC, Pensacola, FL, USA), and active transportation modes of children were indicated by frequency of active trips (FAT) reported by parents.

PA time included the following five components: light PA (LPA), moderate PA (MPA), vigorous PA (VPA), moderate-to-vigorous PA (MVPA), and total PA (TPA). LPA, MPA, and VPA were measured by the accelerometer, whereas MVPA and TPA were obtained by calculation (MVPA = MPA + VPA, TPA = LPA + MPA + VPA). The sampling frequency of the accelerometer was set to 30 Hz. During the test, participants were required to wear accelerometers for seven consecutive days (five weekdays and two weekends) on the right hip [18], and secured by an elastic belt, and required to wear the device at any time except bathing, swimming, and sleeping [19]. Finally, researchers download and analyzed the data using accelerometer analytics software Actilife 6.13.3. The raw data were captured by 1-s sampling interval because of the irregular and intermittent PA of young children [20] based on PA intensity cutoff points developed by Pate et al. (LPA: 150–1680 counts/min, MPA: 1680–3368 counts/min, VPA: >3368 counts/min) [21]. Only data that met PA of at least 8 h per day and 40 min per hour and contained at least 3 days (including one weekend) were considered valid [22].

The FAT for children was obtained by seven multiple-choice questions, the first six of which asked whether children would walk or bike to friends’ home, parks/lawn, public transportation stations, schools, malls, and sports venues. If necessary, the location of question 7 needs to be supplemented by the filler. Each question had the options, not within active transportation distance, never, less than one time a week, 1–2 times a week, 3–4 times a week, 5–6 times a week, and every day, and these options meant scores of 0–6, respectively. If the seventh question was not supplemented, the average value calculated from the non-zero value of the first six questions was the FAT score; otherwise, the average value calculated from the non-zero value of 7 questions was the FAT score. The higher the score, the higher the frequency that children use active transportation modes.

### 2.3. Community Environment

Community environment scores were obtained by parents completing ‘Empirical Study Scale of Physical Activity Environment for Preschool Children’. This scale was revised on several instruments assessing children’s neighborhood environment, and it has acceptable reliability [23,24].

This scale consisted of eight parts. (1) ‘Distance from home to kindergarten’ consisted of a single-choice question with the following four options: within 1 km, 1–3 km, 4–10 km, and more than 10 km, corresponding to 1–4 points. The index score was calculated as an average, and the higher the score, the farther the distance between home and kindergarten. (2) ‘Parental perception of appropriate walking distance’ composed of a fill-in-the-blank question in kilometers, and the score for this indicator was calculated as the average value, with larger values indicating farther distance. (3) ‘Public activity facilities near the community’ consisted of six multiple-choice questions, each representing a facility (park, small square or open space, swimming pool, outdoor sports field, children’s playground, and other places). Each question had two options, a ‘yes’ score of 1 and a ‘no’ score of 0. The score for this indicator was calculated as six plus, and the higher the score, the more public activities near your community. (4) ‘Community transportation environment’ consisted of three single-choice questions (serious traffic congestion, no time to drive children to activities, and limited nearby transportation). (5) ‘Community traffic safety’ consisted of two single-choice questions (worry about road safety and safety hazards caused by too many motor vehicles). (6) ‘Community personal safety’ consisted of three single-choice questions (very worried about strangers, children needing adult supervision for outdoor activities, and children needing adult supervision for activities). (7) ‘Community activity environment’ consisted of four single-choice questions (fewer lights or intersections, dark outdoor environments unsuitable for play in winter, heat outdoors unsuitable for play in summer, and playing without a partner). (8) ‘Convenience of community activities’ consisted of two single-choice questions (children need to cross multiple roads to get to an event site and fewer sports venues near their community). All of the options in the last five parts were divided into ‘totally disagree’, ‘disagree’, ‘not sure’, ‘agree’, and ‘totally agree’ and corresponded to 1–5 points. The scores of the last five parts were calculated by adding the scores of each question to calculate the average value. The higher the score, the worse the environment.

### 2.4. Covariates

Covariates included participants’ ages, heights, body weights, and their parent’s education. Ages of participants and educational levels of their parents were obtained by parental reports. The education levels of parents were classified into the following eight grades: ‘kindergarten degree or below’, ‘primary school education’, ‘junior high school degree’, ‘high school degree’, ‘college degree’, ‘bachelor’s degree’, ‘master’s degree’, ‘doctoral degree’, and corresponded to 1–8 points. The height and weight of the study participants were measured and standardized using a height and weight scale, and body mass index (BMI) was calculated using the formula weight/height^2^ (kg/m^2^) [25].

### 2.5. Statistical Analysis

Statistical analysis was performed using the SPSS version 22.0 (IBM Corp., Armonk, NY, USA). Descriptive information was expressed in percentage or mean ± standard deviation. Independent sample *t*-tests were used to analyze basic characteristics, PA, and community environment for different genders. Paired sample *t*-tests were used to investigate differences in PA between weekdays and weekends. Multivariate linear regression analysis was used to examine the relationship between community environments and PA, with a significance of *p* < 0.05.

## 3. Results

### 3.1. Study Population Characteristics

Table 1 summarizes the characteristics of the study groups. Valid accelerometer data were provided by 304 children (mean age 5.07 ± 0.94 years, 154 boys and 150 girls) and included in the relevant data analysis, with 167 samples excluded due to loss or damage to accelerometers (*n* = 4), invalid data (*n* = 79), or not worn as required (*n* = 84). Independent sample *t*-test results showed no statistical differences between boys and girls in height, weight, BMI, or education level of parents (*p* > 0.05).

### 3.2. Physical Activity

According to China’s first ‘Physical activity guideline for Chinese preschoolers (3–6 years of age) (Expert Consensus Version)’ published in 2018, children should accumulate more than 180 min/day of TPA and should accumulate at least 60 min/day of MVPA [26]. Participants’ PA and meeting guideline recommendations are shown in Table 2.

Children’s FAT scores reported by parents are shown in Table 3. Boys seemed to choose active transportation modes more frequently than girls.

### 3.3. Community Environment

Scores for each community environment indicator reported by parents are shown in Table 4. The first two distance indices were selected as comprehensive variables for the accessibility of activity venues or facilities. The results of an independent sample *t*-test showed no significant difference between boys and girls in all community environment scores (*p* > 0.05).

### 3.4. Multivariate Linear Regression Results of Preschool Children’s Physical Activity and Community Environment

The multivariate linear regression results of children’s different types of PA and community environments are shown in Table 5, Table 6 and Table 7. Results showed that ‘parental perception of appropriate walking distance’ was associated with nearly half indicators of the children’s PA. ‘Public activity facilities near the community’ was associated with FAT for overall children. ‘Community transportation environment’ was associated with overall children’s average day LPA, weekend LPA, MPA, TPA, and FAT for girls. Furthermore, ‘community personal safety’ was associated with boys’ weekday VPA.

## 4. Discussion

This study used the ActigraphGT3X+ accelerometer to objectively measure the PA of preschool children in two population-dense cities in China and examined the relationship between PA and the community environment. The main finding of this study was that parental perceptions of appropriate walking distance, public activity facilities near the community, community transportation environments, and community physical safety were correlated with preschool children’s PA.

### 4.1. Physical Activity Level and Characteristics of Preschooler Children

In this study, only 26.2% of preschool children (33.8% for boys and 18.4% for girls) met China’s TPA and MVPA recommendations [26]. Although the percentage of children meeting MVPA recommendations is high, the TPA has room for improvement. Compared with the results of Zhang and Wang’s studies with large samples of PA in young children [27,28], the proportion of children meeting both the MVPA and TPA on weekdays and weekends was significantly higher in this study. This may be because PA in our study was objectively measured by children wearing accelerometers, whereas the two studies were based on information about participants’ PA obtained from teachers and parents who filled out PA questionnaires, leading to differences in PA measurement. Other studies of children’s PA based on accelerometer data agree with the length of physically active time we recorded in the current study [29,30,31,32].

This study showed that, across all types of PA, children were significantly less physically active on weekdays than on weekends. This is possibly because participants had more leisure time on weekends than on weekdays, leading to an increase in each type of PA. A large proportion of preschool children’s PA is tied to the caregiver’s PA time, and parents have more time off at weekends, and the increase in parental PA is driving children’s PA [32].

This study found significant gender differences in PA between boys and girls as early as childhood. The vast majority of boys in the study were more active than girls on weekends and weekdays. Some researchers found similar gender gaps in their studies [29,32,33], but incongruent findings from current studies persist. For example, Chaput showed that girls had slightly higher LPA, MVPA, and TPA than boys [34], and the reasons should be explored further.

### 4.2. Relationship between Community Environment and Preschool Children’s Physical Activity

This study found ‘parental perception of appropriate walking distance’ was associated with nearly half of the indicators of the children’s PA. The longer a parent considers it appropriate to walk, the more distance they can walk with their child, and the more PA the child will have. This also reflects parents’ perceptions of whether the distance between community activities facilities and houses is appropriate and the importance of reasonable community planning, which could be crucial factors in young children attaining daily PA. Similar to these results, Zhang’s [35] research suggests that mothers’ perceptions of the importance of PA and their daily health behaviors may have a greater impact on their PA than the environment in which their children are physically active. Agard’s results also confirm that social support and active childcare practices, including encouragement, supervision, logistical support, co-participation, and facilitation, are crucial to the improvement of preschool children’s PA [36].

A positive correlation was noted between ‘public activity facilities near the community’ and FAT for overall children, suggesting that the more public activity facilities near the community, the higher the frequency that children use active transportation modes. Roemmich [37] and Lu’s [38] results also confirm this finding. Families living in communities with more activity facilities may have more opportunities to be active, so active parents may push their children to be more active. This study did not find a specific type of PA associated with activity facilities near the community, potentially because the impact between the two was indirect, with intermediate factors such as parents’ perceptions of activity facilities [17].

‘Community transportation environment’ was positively correlated with children’s average day LPA, weekend day LPA, MPA, and TPA and was negatively correlated with FAT for girls. The worse the transportation environment (inconvenient driving or public transportation), the higher the level of PA among children, and the lower the FAT for girls. This may be because the more difficult the transportation environment, the more time parents choose to accompany their children on active journeys (walking or cycling) and ensure their child’s safety. In contrast to boys, girls are less active and therefore less likely to take an active trip in difficult transportation conditions.

‘Community personal safety’ was negatively associated with boys’ weekday VPA, suggesting that higher community personal safety scores (i.e., the less safe) were associated with shorter periods of PA for boys, but not for girls. This could be because boys are generally more active and risk-taking than girls, which causes parents to be concerned. This indicated that parents’ awareness of community personal safety influences their children’s PA [39]. Consistent with this finding are Weir’s results [40]; that is, one of the reasons inner-city children are less active than suburban children is that parents restrict children’s PA for safety reasons.

This study found no significant correlation between ‘distance from home to kindergarten, community activity environment’, ‘convenience of community movement’, and all indicators of PA in preschoolers. Contrary to our findings, Timperio’s study analyzed routes using a digital elevation model based on surface tools and found a negative correlation between distance from home to school and children’s PA [41]. Using Google Maps tools, Terrón-Pérez found that the distance from home to school is a factor in positive child passage [42]. Unlike the two studies that accurately measured distance, this study used subjective questionnaires for distance data collection, which may also be the primary cause of the difference in results. A Dutch study found that the presence of lights or intersections is associated with children’s outdoor activities [43]. This may be due to different population densities and definitions of communities in China and other countries. The subjects in the present study were from Beijing and Zhengzhou, two of the most population-dense cities in China. In these cities, roads and intersections are more crowded and safety concerns are greater, so parents may not allow their children to play in those areas.

### 4.3. Strengths and Limitations

This study has several strengths. Firstly, participants’ PA was objectively measured using accelerometers, which also allowed the researchers to determine PA at various intensity types, enhancing the validity of the external results. Secondly, this research was conducted in a densely populated urban environment, providing an empirical foundation for similar healthy urban construction. Thirdly, some potential confounders were controlled during data analysis. However, this study has some limitations. Firstly, the study is a cross-sectional study, and cause-and-effect relationships between variables are impossible to infer. Secondly, the community environment measurement in this study is in the form of scales for parents of preschool children, and the self-reported survey may have biases. Third, there is an insufficient collection of covariates related to this research. For example, participating families’ purchasing power, parents’ BMI, and their attitude toward PA are also important variables that need to be considered.

## 5. Conclusions

Parental perception of appropriate walking distance, public activity facilities near the community, the community transportation environment, and community personal safety are important factors affecting preschool children’s PA. When planning communities, attention should be paid to appropriate distances between residential areas and public places to increase the number of public facilities for activity and make communities safer, thereby indirectly increasing the PA of preschool children with the help of parents.

Future scholars can use a combination of the global positioning system and accelerometers to measure PA and obtain more information about PA locations to facilitate a visual understanding of the characteristics of children’s PA. They can also incorporate different scientific field instruments into research, such as using an environmental audit tool integrated with a geographic information system when measuring community environmental indicators. To make up for the deficiency that the questionnaire collection is more subjective and does not allow for the personalized collection of environmental characteristics.

## Figures and Tables

**Table 1 ijerph-19-14797-t001:** Participants’ demographic characteristics.

Variable	Total	Boys	Girls
N	304	154	150
Age (years)	5.07 ± 0.94	5.11 ± 0.96	5.02 ± 0.91
Height (cm)	110.53 ± 7.44	110.85 ± 7.23	110.28 ± 7.65
Weight (kg)	19.87 ± 3.92	20.05 ± 3.62	19.67 ± 3.90
BMI (kg/m^2^)	16.15 ± 1.81	16.22 ± 1.77	16.05 ± 1.64
Level of education of parents (1–8 points)	5.72 ± 0.97	5.78 ± 0.93	5.66 ± 1.02

**Table 2 ijerph-19-14797-t002:** Participants’ PA time and meeting guideline recommendations.

Variable	Total	Boys	Girls
Average day (min/day)			
LPA	95.42 ± 15.84	97.33 ± 16.83 *	93.28 ± 14.56
MPA	42.64 ± 9.96	44.90 ± 10.97 *	40.10 ± 7.93
VPA	35.08 ± 11.56	38.11 ± 11.99 *	32.17 ± 10.00
MVPA	77.58 ± 18.78	83.01 ± 19.31 *	72.00 ± 16.71
TPA	173.26 ± 30.38	180.34 ± 32.54 *	165.77 ± 26.30
Meeting MVPA recommendations (%)	82.8	90.3	74.8
Meeting TPA recommendations (%)	38.3	50.0	26.0
Meeting MVPA and TPA recommendations (%)	26.2	33.8	18.4
Weekday (min/day)			
LPA	93.47 ± 15.85 #	95.07 ± 17.07 #	91.59 ± 14.40 #
MPA	40.99 ± 10.13 #	42.98 ± 11.62 *#	38.74 ± 7.68 #
VPA	34.60 ± 11.58 #	37.29 ± 12.23 *#	31.98 ± 9.83 #
MVPA	75.59 ± 18.37 #	80.27 ± 19.74 *#	70.72 ± 15.50 #
TPA	169.06 ± 30.86 #	175.34 ± 33.61 *#	162.31 ± 26.33 #
Meeting MVPA recommendations (%)	81.2	86.4	75.3
Meeting TPA recommendations (%)	33.8	41.6	26.0
Meeting MVPA and TPA recommendations (%)	23.1	28.1	18.4
Weekend (min/day)			
LPA	101.17 ± 24.75	103.41 ± 25.43	98.75 ± 23.90
MPA	47.86 ± 16.24	51.00 ± 17.58 *	44.33 ± 13.79
VPA	36.92 ± 16.19	40.13 ± 16.99 *	33.89 ± 14.60
MVPA	84.78 ± 29.30	91.12 ± 31.02 *	78.22 ± 26.35
TPA	185.95 ± 48.81	194.54 ± 51.30 *	176.97 ± 45.00
Meeting MVPA recommendations (%)	80.2	87	72.7
Meeting TPA recommendations (%)	56.2	62.3	49.3
Meeting MVPA and TPA recommendations (%)	38.1	42.1	34.4

LPA: light PA; MPA: moderate PA; VPA: vigorous PA; MVPA: moderate-to-vigorous PA; TPA: total PA; * *p* < 0.05 compared with girls; # *p* < 0.05 compared with weekends.

**Table 3 ijerph-19-14797-t003:** FAT scores of participants.

Variable	Total	Boys	Girls
FAT score	2.50 ± 0.68	2.53 ± 0.72	2.47 ± 0.64
Friend’s home (0–6 points)	2.18 ± 1.01	2.23 ± 1.09	2.14 ± 0.93
Parks/lawns (0–6 points)	2.89 ± 1.01	3.00 ± 1.12 *	2.78 ± 0.88
Public transportation stations (0–6 points)	2.28 ± 1.13	2.42 ± 1.25 *	2.13 ± 0.98
Schools (0–6 points)	3.49 ± 1.79	3.42 ± 1.83	3.55 ± 1.74
Malls (0–6 points)	2.48 ± 0.95	2.45 ± 0.95	2.51 ± 0.95
Sports venues (e.g., tennis courts, swimming pools, etc.) (0–6 points)	1.91 ± 0.92	1.97 ± 0.97	1.84 ± 0.87
Other places (please specify) (0–6 points)	2.18 ± 1.25	2.00 ± 0.82	2.29 ±1.50

* *p* < 0.05 compared with girls.

**Table 4 ijerph-19-14797-t004:** Community environment scores of participants.

Variable	Total	Boys	Girls
Distance from home to kindergarten (1–4 points)	1.68 ± 0.79	1.69 ± 0.78	1.66 ± 0.80
Parental perception of appropriate walking distance (0.5–5 km)	1.44 ± 1.00	1.52 ± 1.20	1.36 ± 0.73
Public activity facilities near the community (0–6 points)	4.83 ± 1.25	4.90 ± 1.20	4.76 ± 1.29
Community transportation environment (1–5 points)	2.62 ± 0.72	2.65 ± 0.75	2.59 ± 0.69
Community traffic safety (1–5 points)	3.57 ± 0.93	3.57 ± 0.93	3.57 ± 0.93
Community personal safety (1–5 points)	3.74 ± 0.85	3.75 ± 0.81	3.74 ± 0.90
Community activity environment (1–5 points)	2.89 ± 0.74	2.94 ± 0.71	2.84 ± 0.77
Convenience of community activities (1–5 points)	2.79 ± 0.87	2.84 ± 0.89	2.76 ± 0.85

**Table 5 ijerph-19-14797-t005:** Multivariate linear regression analysis of community environment and average day physical activity of participants (*B*, 95% CI).

Variable	LPA	MPA	VPA	MVPA	TPA	FAT
Total						
Distance from home to kindergarten	1.494 (−1.376, 4.364)	1.299 (−0.296, 2.894)	0.836 (−1.121, 2.792)	2.314 (−1.002, 5.629)	3.492 (−2.003, 8.986)	−0.039 (−0.153, 0.076)
Parental perception of appropriate walking distance	0.941 (−1.992, 3.875)	1.009 (−0.621, 2.639)	2.682 (0.685, 4.680) *	3.948 (0.563, 7.332) *	4.436 (−1.181, 10.052)	0.007 (−0.118, 0.132)
Public activity facilities near the community	0.206 (−1.791, 2.203)	−0.541 (−1.650, 0.569)	0.551 (−0.812, 1.913)	0.064 (−2.245, 2.373)	0.174 (−3.649, 3.998)	0.099 (0.014, 0.183) *
Community transportation environment	4.034 (0.012, 8.056) *	1.833 (−0.402, 4.068)	0.650 (−2.095, 3.396)	2.542 (−2.111, 7.194)	6.473 (−1.228, 14.173)	−0.111 (−0.270, 0.048)
Community traffic safety	−1.110 (−4.027, 1.807)	−0.749 (−2.370, 0.872)	−0.037 (−2.029, 1.955)	−0.813 (−4.188, 2.562)	−1.875 (−7.461, 3.711)	−0.052 (−0.168, 0.064)
Community personal safety	−0.489 (−3.520, 2.543)	0.217 (−1.468, 1.901)	−0.829 (−2.875, 1.217)	−0.058 (−3.524, 3.408)	−1.525 (−7.330, 4.279)	0.081 (−0.043, 0.205)
Community activity environment	−0.529 (−4.537, 3.479)	−0.112 (−2.339, 2.115)	1.885 (−0.837, 4.606)	2.270 (−2.342, 6.882)	0.862 (−6.812, 8.536)	−0.011 (−0.170, 0.148)
Convenience of community activities	−1.500 (−4.997, 1.998)	−0.194 (−2.137, 1.750)	−1.384 (−3.695, 0.927)	−2.635 (−6.551, 1.280)	−2.266 (−8.963, 4.430)	0.112 (−0.021, 0.245)
Boys						
Distance from home to kindergarten	0.726 (−3.837, 5.290)	1.475 (−0.782, 3.731)	0.693 (−1.921, 3.307)	2.168 (−2.315, 6.650)	2.894 (−5.290, 11.078)	0.021 (−0.165, 0.208)
Parental perception of appropriate walking distance	0.641 (−3.738, 5.021)	0.013 (−2.153, 2.178)	2.063 (−0.445, 4.572)	2.076 (−2.225, 6.378)	2.718 (−5.136, 10.571)	−0.027 (−0.228, 0.174)
Public activity facilities near the community	0.017 (−3.379, 3.413)	−0.641 (−2.321, 1.038)	0.343 (−1.603, 2.288)	−0.299 (−3.635, 3.307)	−0.282 (−6.372, 5.809)	0.100 (−0.047, 0.247)
Community transportation environment	4.002 (−1.932, 9.936)	0.657 (−2.277, 3.591)	−0.127 (−3.526, 3.272)	0.530 (−5.299, 6.359)	4.532 (−6.110, 15.173)	−0.023 (−0.263, 0.217)
Community traffic safety	−1.035 (−5.592, 3.521)	−0.853 (−3.106, 1.400)	0.141 (−2.469, 2.751)	−0.712 (−5.188, 3.764)	−1.747 (−9.919, 6.424)	−0.134 (−0.320, 0.052)
Community personal safety	−0.950 (−5.582, 3.683)	−0.487 (−2.778, 1.803)	−2.199 (−4.852, 0.455)	−2.686 (−7.237, 1.864)	−3.636 (−11.943, 4.672)	0.022 (−0.185, 0.229)
Community activity environment	1.800 (−4.373, 7.974)	1.566 (−1.487, 4.619)	2.206 (−1.331, 5.742)	3.772 (−2.292, 9.836)	5.572 (−5.499, 16.644)	0.045 (−0.214, 0.303)
Convenience of community activities	−1.589 (−7.144, 3.967)	−0.750 (−3.497, 1.997)	−0.619 (−3.801, 2.564)	−1.369 (−6.826, 4.088)	−2.957 (−12.920, 7.005)	0.190 (−0.033, 0.413)
Girls						
Distance from home to kindergarten	0.544 (−3.269, 4.356)	−0.255 (−2.479, 1.968)	−0.461 (−3.380, 2.458)	−0.128 (−4.872, 4.615)	−0.593 (−7.909, 6.723)	−0.106 (−0.255, 0.043)
Parental perception of appropriate walking distance	1.328 (−2.839, 5.495)	3.261 (0.831, 5.691) *	4.212 (1.010, 7.415) *	7.915 (2.711, 13.120) *	8.486 (0.490, 16.483) *	0.036 (−0.126, 0.199)
Public activity facilities near the community	−0.239 (−2.638, 2.159)	−1.085 (−2.484, 0.314)	−0.078 (−1.927, 1.772)	−1.070 (−4.075, 1.935)	−1.469 (−6.072, 3.134)	0.082 (−0.019, 0.183)
Community transportation environment	2.992 (−3.031, 9.015)	2.761 (−0.751, 6.274)	0.596 (−4.039, 5.232)	3.843 (−3.690, 11.376)	6.003 (−5.555, 17.560)	−0.223 (−0.443, −0.003) *
Community traffic safety	−0.937 (−4.812, 2.938)	−0.151 (−2.411, 2.108)	−0.060 (−3.044, 2.924)	−0.469 (−5.318, 4.380)	−0.964 (−8.399, 6.472)	0.050 (−0.098, 0.199)
Community personal safety	0.012 (−4.234, 4.258)	1.753 (−0.723, 4.229)	0.932 (−2.253, 4.117)	3.953 (−1.224, 9.129)	1.791 (−6.357, 9.939)	0.090 (−0.066, 0.246)
Community activity environment	−2.891 (−8.239, 2.458)	−2.409 (−5.528, 0.710)	1.216 (−2.885, 5.317)	−0.457 (−7.121, 6.208)	−4.610 (−14.873, 5.653)	−0.059 (−0.260, 0.142)
Convenience of community activities	−1.351 (−5.940, 3.237)	0.178 (−2.498, 2.854)	−1.912 (−5.252, 1.427)	−3.691 (−9.118, 1.736)	−1.688 (−10.494, 7.118)	0.082 (−0.085, 0.249)

*B*, non-standardized regression coefficient; * *p* < 0.05; multivariate linear regression control variables were age and BMI of the participants and education level of their parents.

**Table 6 ijerph-19-14797-t006:** Multivariate linear regression analysis of community environment and weekday physical activity of participants (*B*, 95% CI).

Variable	LPA	MPA	VPA	MVPA	TPA
Total					
Distance from home to kindergarten	1.974 (−0.915, 4.864)	1.425 (−0.187, 3.036)	0.814 (−1.195, 2.823)	2.239 (−1.058, 5.536)	4.213 (−1.443, 9.869)
Parental perception of appropriate walking distance	0.801 (−2.153, 3.755)	0.741 (−0.906, 2.388)	2.138 (0.084, 4.192) *	2.879 (−0.491, 6.249)	3.680 (−2.102, 9.462)
Public activity facilities near the community	0.203 (−1.808, 2.213)	−0.457 (−1.578, 0.665)	0.343 (−1.055, 1.741)	−0.113 (−2.407, 2.181)	0.090 (−3.846, 4.025)
Community transportation environment	2.985 (−1.065, 7.034)	1.262 (−0.996, 3.521)	0.311 (−2.505, 3.127)	1.574 (−3.047, 6.194)	4.558 (−3.368, 12.485)
Community traffic safety	−0.514 (−3.451, 2.424)	−0.169 (−1.807, 1.469)	0.297 (−1.745, 2.340)	0.129 (−3.223, 3.480)	−0.385 (−6.135, 5.365)
Community personal safety	−1.366 (−4.418, 1.687)	−0.360 (−2.062, 1.342)	−1.916 (−4.039, 0.206)	−2.276 (−5.759, 1.207)	−3.642 (−9.617, 2.333)
Community activity environment	0.737 (−3.299, 4.772)	0.783 (−1.468, 3.033)	2.286 (−0.520, 5.092)	3.069 (−1.536, 7.673)	3.805 (−4.094, 11.705)
Convenience of community activities	−1.717 (−5.239, 1.804)	−0.506 (−2.470, 1.458)	−1.225 (−3.674, 1.224)	−1.731 (−5.750, 2.287)	−3.448 (−10.342, 3.445)
Boys					
Distance from home to kindergarten	1.441 (−3.226, 6.108)	1.796 (−0.613, 4.205)	1.447 (−1.443, 4.336)	3.243 (−1.675, 8.160)	4.683 (−4.080, 13.447)
Parental perception of appropriate walking distance	−0.148 (−4.627, 4.331)	−0.547 (−2.859, 1.764)	1.100 (−1.673, 3.873)	0.553 (−4.166, 5.272)	0.404 (−8.005, 8.814)
Public activity facilities near the community	0.079 (−3.394, 3.553)	−0.614 (−2.406, 1.179)	0.238 (−1.912, 2.389)	−0.375 (−4.035, 3.284)	−0.296 (−6.818, 6.226)
Community transportation environment	3.745 (−2.324, 9.814)	0.550 (−2.582, 3.682)	0.026 (−3.732, 3.783)	0.575 (−5.819, 6.969)	4.321 (−7.074, 15.715)
Community traffic safety	−0.897 (−5.557, 3.763)	−0.437 (−2.842, 1.968)	0.216 (−2.669, 3.101)	−0.221 (−5.131, 4.689)	−1.118 (−9.868, 7.632)
Community personal safety	−2.306 (−7.044, 2.432)	−1.205 (−3.650, 1.241)	−3.012 (−5.946, −0.079) *	−4.217 (−9.209, 0.775)	−6.523 (−15.419, 2.373)
Community activity environment	3.629 (−2.684, 9.943)	2.549 (−0.710, 5.808)	2.978 (−0.932, 6.887)	5.527 (−1.126, 12.179)	9.156 (−2.699, 21.011)
Convenience of community activities	−1.340 (−7.021, 4.342)	−0.815 (−3.747, 2.118)	−0.763 (−4.280, 2.755)	−1.577 (−7.564, 4.409)	−2.917 (−13.585, 7.751)
Girls					
Distance from home to kindergarten	0.750 (−3.008, 4.508)	−0.279 (−2.414, 1.857)	−1.408 (−4.273, 1.456)	−1.687 (−6.002, 2.628)	−0.937 (−8.203, 6.329)
Parental perception of appropriate walking distance	1.635 (−2.472, 5.742)	3.062 (0.728, 5.396) *	4.075 (0.944, 7.206) *	7.137 (2.420, 11.853) *	8.772 (0.830, 16.714) *
Public activity facilities near the community	−0.179 (−2.544, 2.185)	−0.874 (−2.218, 0.469)	−0.353 (−2.156, 1.449)	−1.227 (−3.943, 1.488)	−1.407 (−5.979, 3.165)
Community transportation environment	1.656 (−4.280, 7.593)	1.934 (−1.439, 5.307)	−0.341 (−4.886, 4.185)	1.594 (−5.223, 8.410)	3.250 (−8.229, 14.729)
Community traffic safety	0.484 (−3.336, 4.303)	0.747 (−1.423, 2.917)	0.971 (−1.941, 3.882)	1.718 (−2.668, 6.103)	2.201 (−5.184, 9.586)
Community personal safety	0.278 (−3.907, 4.463)	1.607 (−0.771, 3.985)	−0.162 (−3.352, 3.028)	1.445 (−3.360, 6.251)	1.724 (−6.368, 9.816)
Community activity environment	−2.997 (−6.326, 2.720)	−2.016 (−5.011, 0.979)	0.984 (−3.034, 5.003)	−1.032 (−7.085, 5.022)	−4.028 (−14.221, 6.165)
Convenience of community activities	−1.803 (−6.326, 2.720)	−0.205 (−2.775, 2.365)	−1.345 (−4.793, 2.103)	−1.550 (−6.744, 3.644)	−3.353 (−12.099, 5.392)

*B*, non-standardized regression coefficient; * *p* < 0.05; multivariate linear regression control variables were age and BMI of the participants and education level of their parents.

**Table 7 ijerph-19-14797-t007:** Multivariate linear regression analysis of community environment and weekend physical activity of participants (*B*, 95% CI).

Variable	LPA	MPA	VPA	MVPA	TPA
Total					
Distance from home to kindergarten	0.220 (−4.331, 4.772)	1.087 (−1.676, 3.850)	0.333 (−2.564, 3.2230)	1.420 (−3.836, 6.675)	1.640 (−7.384, 10.664)
Parental perception of appropriate walking distance	2.633 (−2.020, 7.286)	2.295 (−0.529, 5.120)	4.007 (1.045, 6.968) *	6.302 (0.930, 11.674) *	8.935 (−0.290, 18.159)
Public activity facilities near the community	0.464 (−2.703, 3.631)	−0.655 (−2.577, 1.268)	0.987 (−1.029, 3.003)	0.332 (−3.324, 3.989)	0.797 (−5.482, 7.076)
Community transportation environment	8.278 (1.900, 14.657) *	4.485 (0.613, 8.357) *	2.014 (−2.046, 6.074)	6.499 (−0.867, 13.864)	14.777 (2.130, 27.424) *
Community traffic safety	−2.391 (−7.018, 2.236)	−2.439 (−5.248, 0.370)	−0.743 (−3.688, 2.203)	−3.181 (−8.524, 2.161)	−5.572 (−14.746, 3.602)
Community personal safety	2.332 (−2.476, 7.140)	1.692 (−1.227, 4.611)	0.401 (−2.660, 3.461)	2.093 (−3.459, 7.644)	4.425 (−5.108, 13.958)
Community activity environment	−4.640 (−10.997, 1.717)	−2.442 (−6.301, 1.416)	−0.412 (−4.458, 3.634)	−2.854 (−10.194, 4.486)	−7.495 (−20.098, 5.109)
Convenience of community activities	−2.246 (−7.793, 3.301)	0.206 (−3.162, 3.573)	0.876 (−2.655, 4.407)	1.082 (−5.324, 7.487)	−1.165 (−12.163, 9.834)
Boys					
Distance from home to kindergarten	−1.726 (−8.639, 5.186)	0.428 (−3.705, 4.562)	−1.539 (−5.472, 2.393)	−1.111 (−8.539, 6.317)	−2.837 (−16.068, 10.394)
Parental perception of appropriate walking distance	4.554 (−2.079, 11.188)	2.413 (−1.554, 6.381)	5.120 (1.346, 8.894) *	7.533 (0.405, 14.662) *	12.088 (−0.610, 24.785)
Public activity facilities near the community	−0.594 (−5.739, 4.550)	−1.327 (−4.404, 1.749)	0.382 (−2.544, 3.309)	−0.945 (−6.473, 4.583)	−1.539 (−11.387, 8.308)
Community transportation environment	5.195 (−3.793, 14.183)	1.562 (−3.814, 6.937)	−0.153 (−5.266, 4.960)	1.409 (−8.251, 11.068)	6.603 (−10.601, 23.808)
Community traffic safety	−1.200 (−8.102, 5.702)	−2.777 (−6.904, 1.351)	−0.104 (−4.031, 3.822)	−2.881 (−10.298, 4.536)	−4.081 (−17.292, 9.130)
Community personal safety	2.480 (−4.537, 9.497)	1.352 (−2.845, 5.548)	−0.540 (−4.532, 3.452)	0.811 (−6.730, 8.352)	3.291 (−10.140, 16.723)
Community activity environment	−2.877 (−12.229, 6.474)	−0.577 (−6.170, 5.015)	0.887 (−4.433, 6.207)	0.310 (−9.740, 10.359)	−2.568 (−20.467, 15.332)
Convenience of community activities	−4.804 (−13.219, 3.611)	−2.347 (−7.379, 2.686)	−1.542 (−6.329, 3.245)	−3.889 (−12.932, 5.154)	−8.693 (−24.800, 7.415)
Girls					
Distance from home to kindergarten	0.747 (−5.798, 7.293)	0.038 (−3.777, 3.852)	0.858 (−3.600, 5.317)	0.896 (−6.755, 8.547)	1.643 (−11.212, 14.498)
Parental perception of appropriate walking distance	2.194 (−4.960, 9.348)	4.808 (0.639, 8.977) *	4.845 (−0.028, 9.718)	9.653 (1.291, 18.016) *	11.847 (−2.203, 25.898)
Public activity facilities near the community	0.441 (−3.677, 4.560)	−1.187 (−3.587, 1.213)	0.670 (−2.135, 3.476)	−0.517 (−5.331, 4.298)	−0.075 (−8.164, 8.013)
Community transportation environment	9.421 (−0.919, 19.761)	5.699 (−0.326, 11.725)	2.722 (−4.321, 9.765)	8.422 (−3.665, 20.508)	17.842 (−2.465, 38.150)
Community traffic safety	−4.357 (−11.009, 2.295)	−1.809 (−5.685, 2.068)	−1.692 (−6.223, 2.839)	−3.501 (−11.277, 4.275)	−7.858 (−20.923, 5.207)
Community personal safety	0.324 (−6.965, 7.613)	2.017 (−2.231, 6.264)	0.781 (−4.185, 5.746)	2.797 (−5.724, 11.318)	3.121 (−11.195, 17.438)
Community activity environment	−3.876 (−13.057, 5.306)	−3.676 (−9.027, 1.674)	−0.225 (−6.479, 6.029)	−3.901 (−14.635, 6.832)	−7.777 (−25.810, 10.256)
Convenience of community activities	−1.095 (−8.973, 6.783)	2.031 (−2.560, 6.622)	2.127 (−3.240, 7.493)	4.157 (−5.052, 13.367)	3.063 (−12.410, 18.535)

*B*, non-standardized regression coefficient; * *p* < 0.05; multivariate linear regression control variables were age and BMI of the participants and education level of their parents.

## Data Availability

The data presented in this study are available on request from the corresponding author. The data are not publicly available due to privacy restrictions.

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
