# Peer review of "Preschool Children’s Physical Activity and Community Environment: A Cross-Sectional Study of Two Cities in China"

_ijerph, 2022, doi:10.3390/ijerph192214797_

Round 1

Reviewer 1 Report

First of all I would like to thank the authors for the possibility to review this work. I really think it is a very important work on children's health. As the authors point out, obesity is a public health problem not only in China, but also in more and more countries with this problem in the paediatric population. That is why I think this is a very good research. Similarly, research into the different factors that may influence the practice of regular physical activity, such as children's community environment, is essential for possible policy measures to encourage higher rates of physical activity in children, not just preschoolers.

The article is of the necessary quality to be published in this journal, being of high quality in all its sections. It has a solid theoretical and scientific argumentation, with a clear methodology and a very exhaustive analysis. Likewise, the discussion and conclusion are of high quality. I would only suggest for future research, to take into account a greater number of covariates that may be of interest for the research. In this case, the purchasing power of the participating families is also an important covariate to take into account.

All in all, congratulations for the research and for the brilliance with which it has been captured in this article.

Reviewer 2 Report

After incorporating my notes, I recommend the paper for publishing. 

Reviewer 3 Report

China is one of the most populous countries in the world, with one of the fastest-growing economies. However, its main metropolises are suffering from overcrowding problems and changing lifestyles. The above puts the health of future generations at risk. Studies about the effect of growth and its consequences on social, community, and family behavior are essential. In this sense, the present work studies, in a transversal way, the possible association of the environment of two dense cities on the degree of participation in physical activities in preschool children. Physical activity was measured by validated actigraphy and parent interviews, and community environment by questionnaires and parent interviews.

As the authors mention in limitations, the study of the environment should be evaluated by objective and more reliable methods. Hence the current need to incorporate different areas of science into studies and for educational authorities to make tools and resources available to researchers so that their results and conclusions apply to other populations. Instruments that are already available in engineering areas.

My opinion is that the manuscript is well written. The introduction is concise, and enough to present the problem's importance and the work's justification. The complete methods are well described and with the appropriate statistical analyses, although, as already mentioned, the measurement instruments are insufficient. Tables 5-6 could be separated from the manuscript, presented in annexes, or sent after the references at the end of the document. The conclusion is according to the title of the work and objectives. Recommendations to improve the study, relevant. The references are current and relevant.

Reviewer 4 Report

Overall, a very strong paper.  I believe it adds to the literature in this topic. I have mostly small edits to pass on.

Line 38 One of the main reasons for overweight and obesity in children is physical 38 inactivity [4]

Recommend rephrasing to “one of the factors associated with raising rates of obesity in a decrease in PA”

Line 58 However, the findings of the existing studies are controversial and inconclusive as they are inconsistent.

Are they controversial or just inconsistent and inconclusive?

Line 63

be investigated to acquire sufficient information to guide intervention plans. Intervention plans around urban planning or around PA strategies for children? Somewhat unclear.

line 77

 active trips—are you referring to active transportation modes?  If so, it might be helpful to delineate that here. It does come up at the end of the paragraph and clarifies it there.

Line 85 I understand that you used accelerometers to capture movement data and used Pate’s formula to delineate between light-moderate and vigorous movement. Is that all you did with the data? I am thinking about helping others to reproduce this experiment. Other authors do provide formulas or more explanation of exactly what the software does with the data and that may be appropriate here.

Line 137

It would have been interesting to get the height and weight of the parents too to be able to see any correlations between parental weight, children’s weight, and beliefs about active transportation.

Table 3. I am confused if the data representing the frequency of active trips, which is what the table is labeled, or are they the mean scores between 0-6 that parents reported?

Same with Table 4. The t-test results are for active transport or for the parental rating?

Line. 178 Section 3.4 My preference is to have results reported and not interpreted until the discussion section. Identified significant p-values should be sufficient in this section for readers to interpret the table. This is a relatively easy fix. 

Line 233 In this study, only 26.2% of preschool children (33.8% for boys and 18.4% for girls) met both recommendations.

Please be specific about which recommendations, since you have data for several categories.

Line 236

 ...the proportion of both MVPA and TPA meeting on weekdays and weekends was significantly higher in this study.

Suggest: ...the proportion of children meeting both the MVP and TPA on weekdays and weekends was significantly higher in this study.

Line 241 Other studies of children’s PA based on accelerometers show that children are physically active for close to as long as our study showed.

Suggest: Other studies of children’s PA based on accelerometer data agree with the length of physically active time we recorded in the current study.

Line 249  Suggest that the rest of this paragraph be a new paragraph since it switches to discussing the differences between boys and girls.

Line 251

Some researchers found a gender gap in their study  

Suggest: Some researchers found similar gender gaps in their results (or their studies)

Line 259

This also reflects from the side that whether the distance between the community activity facilities and the house is appropriate is crucial to children’s PA.

Suggest: This also reflects on the parental belief on whether the distance between the community activities facilities and the house is appropriate, which could be a crucial factor in young children attaining daily PA.

I think you could also link this to community planning…?

Line 280 I think it is very interesting that parents were more concerned about personal safety of boys than girls. I think the inverse would be true in the US. Could this also be associated with risk-taking behaviors in young boys?

Line 296 A Dutch study finds that the presence of lights or inter..

A Dutch study found that the presence…
